# A Careful Examination of Large Language Model Performance on Grade School Arithmetic

**Hugh Zhang**[*]   **Jeff Da**   **Dean Lee**   **Vaughn Robinson**   **Catherine Wu**   **Will Song**

**Tiffany Zhao**   **Pranav Raja**   **Charlotte Zhuang**   **Dylan Slack**   **Qin Lyu**

**Sean Hendryx**   **Russell Kaplan**   **Michele (Mike) Lunati**[†]   **Summer Yue**[†]

Scale AI

## Abstract

Large language models (LLMs) have achieved impressive success on many benchmarks for mathematical reasoning. However, there is growing concern that some of this performance actually reflects dataset contamination, where data closely resembling benchmark questions leaks into the training data, instead of true reasoning ability. To investigate this claim rigorously, we commission *Grade School Math 1000* (GSM1k). GSM1k is designed to mirror the style and complexity of the established GSM8k benchmark, the gold standard for measuring elementary mathematical reasoning. We ensure that the two benchmarks are comparable across important metrics such as human solve rates, number of steps in solution, answer magnitude, and more. When evaluating leading open- and closed-source LLMs on GSM1k, we observe accuracy drops of up to 8%, with several families of models showing evidence of systematic overfitting across almost all model sizes. Further analysis suggests a positive relationship (Spearman's $r^2 = 0.36$) between a model's probability of generating an example from GSM8k and its performance gap between GSM8k and GSM1k, suggesting that some models may have partially memorized GSM8k. Nevertheless, many models, especially those on the frontier, show minimal signs of overfitting, and all models broadly demonstrate generalization to novel math problems guaranteed to not be in their training data.

## 1   Introduction

Improving reasoning in large language models (LLMs) is one of the most important directions of current research. As such, proper benchmarking of current LLM abilities is paramount for ensuring progress continues in the correct direction. Currently, the field typically relies on public benchmarks such as GSM8k (Cobbe et al. [2021]), MATH (Hendrycks et al. [2021b]), MBPP (Austin et al. [2021]), HumanEval (Chen et al. [2021]), SWEBench (Jimenez et al. [2024])). However, because LLMs are trained on large corpora of data scraped from the Internet, there are major concerns

---

[*]Correspondence to hugh.zhang@scale.com   [†]equal senior authorship

Submitted to the 38th Conference on Neural Information Processing Systems (NeurIPS 2024) Track on Datasets and Benchmarks. Do not distribute.

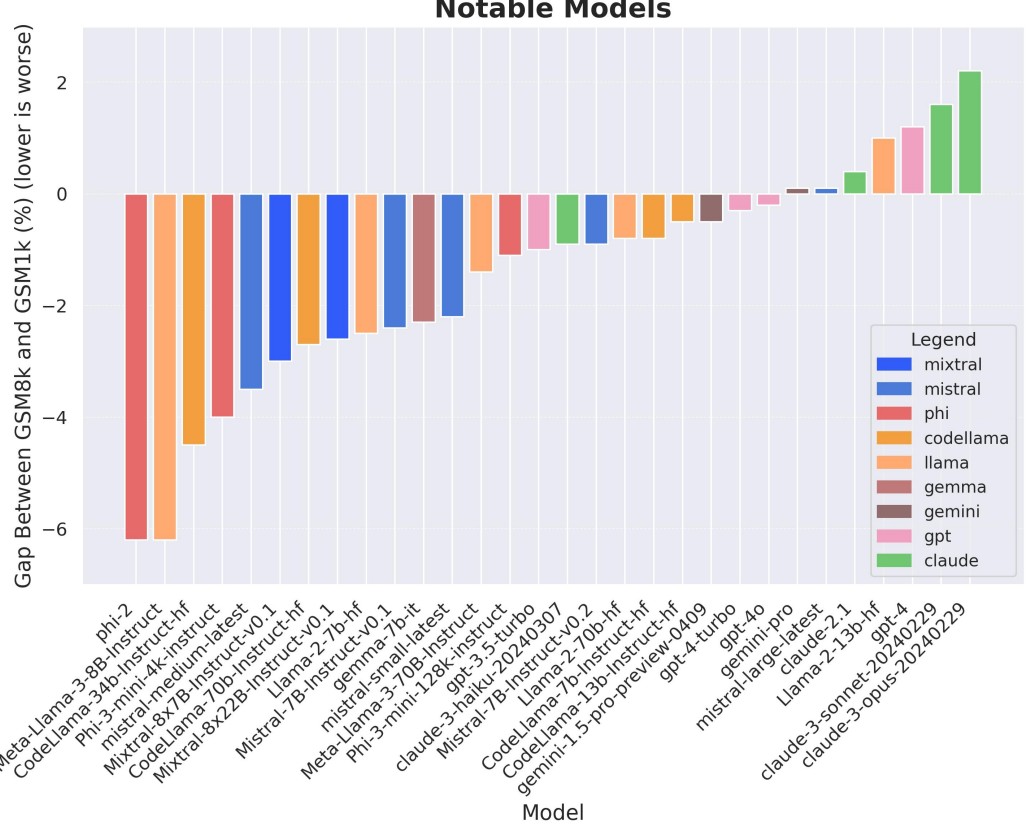

Figure 1: Notable models arranged by their drop in performance between GSM8k and GSM1k (lower is worse). We notice that Phi, Mistral and some models in the Llama family seem to be overfitting GSM8k, while models such as Gemini, GPT, and Claude show little to no signs of overfitting.

that such benchmarks may inadvertently include examples that closely resemble the questions found in such benchmarks. This contamination may result in models having weaker reasoning capabilities than otherwise believed, due to simply being able to repeat the correct answer that it previously encountered during pre- or post- training. To properly investigate the reasoning abilities of models, we commission GSM1k, a newly constructed collection of 1205 grade school level math problems designed to mirror that of GSM8k. We take extensive efforts to ensure that GSM1k has a similar distribution of difficulty to GSM8k to ensure an apples-to-apples comparison. These efforts are described in Section 3, alongside a detailed description of the data creation process. To mitigate worries about data contamination, we created GSM1k solely with human annotators, without assistance from any LLM or other synthetic data source.

| Dataset | Example |
|---|---|
| GSM8k | James writes a 3-page letter to 2 different friends twice a week. How many pages does he write a year? |
| GSM1k (ours) | Lee bought 6 shares of Delta stock at $40 per share. If he wants to make $24 from this trade, how much should Delta stock be per share when he sells? |

Figure 2: Example from both the GSM8k dataset and the new GSM1k dataset (ours). We also provide an additional 50 examples from GSM1k in Appendix E.

We benchmark leading open- and closed-source LLMs on GSM1k, including GPT-4 (OpenAI et al. [2024]), Gemini (Team et al. [2024]), Claude, Mistral (Jiang et al. [2024, 2023]), Llama (Touvron et al. [2023a,b]), Phi (Gunasekar et al. [2023], Abdin et al. [2024]) and many more. Our analysis confirms the widespread suspicion in the field that many models are contaminated by benchmark data, with the worst models performing 8% worse on GSM1k compared to GSM8k. Additionally, our results suggest that several families of models show consistent evidence of overfitting for nearly all model versions and sizes. Further analysis finds a positive relationship (Spearman's $r^2 = 0.36$) between a model's probability of generating examples from GSM8k and its performance gap between GSM8k and GSM1k, strongly suggesting that one important component of this overfitting is that models have partially memorized examples from GSM8k. Nevertheless, our results find that all frontier models show minimal signs of overfitting. Additionally, we also find that all models, including the most overfit ones, are still capable of successfully generalizing to new mathematical grade school problems, albeit occasionally at lower rates than their benchmark numbers would suggest.

We do not intend to release GSM1k publicly at this time to prevent a similar problem of data contamination occurring in the future. However, we plan to run recurring evaluations of all major open- and closed- source releases and to continually update our results. We will also open source our entire evaluation code so that the public version of our results can be reproduced. Additionally, we commit to open sourcing the entire benchmark when either 1) the top open source models score over 95% on GSM1k or 2) June 2025, whichever comes earlier. See Section 3 for precise release criteria.

# 2 Related Work

A major inspiration of this work was the celebrated study on overfitting done on ImageNet classifiers in 2019 (Recht et al. [2019]). This work measured overfitting in ImageNet by creating new versions of CIFAR10 and ImageNet and measuring the performance gap between the public test set and the newly created sets they constructed. In this work, we do a similar analysis on GSM8k, one of the leading benchmarks for mathematical reasoning. GSM1k is modelled after the GSM8k dataset (Cobbe et al. [2021]), released by OpenAI in 2021, which consists of 8.5k grade school math problems. Each problem is designed to be solvable using only basic arithmetic operations ($+$, $-$, $\times$, $\div$) with a difficulty level appropriate for grade school students. As of June 2024, top models report benchmark accuracies of over 95% (Team et al. [2024]). Other popular benchmarks for reasoning include MATH (Hendrycks et al. [2021b]) , MMLU (Hendrycks et al. [2021a]), GPQA (Rein et al. [2023]).

## 2.1 Data Contamination

Because data contamination is a well known issue in the field (Balloccu et al. [2024], Magar and Schwartz [2022], Sainz et al. [2023], Jacovi et al. [2023], Xu et al. [2024]), model builders will frequently take great pains to minimize the likelihood of data contamination. For example, it is common to remove all data with too high of an n-gram overlap with the benchmark data (Brown et al. [2020]). Additionally, methods such as using embedding similarity attempt to remove all contaminated data that is too similar in embedding space to the dataset (Shi et al. [2024]).

Xu et al. [2024] propose using similar variants of a benchmark questions to detect if models favor the original wording as a proxy for data contamination. Srivastava et al. [2024] propose functional evaluations, where benchmarks are written in the form of functions that can generate an infinite number of specific evaluation datapoints, each with slightly different numbers. In this setup, whenever a language model is evaluated, functional evaluations generate a specific problem instance to evaluate the model on, which is then never used again. This reduces the worry of data contamination by ensuring that no datapoint is ever used twice. Like ours, their results indicate the LLMs may be severely overfit on benchmark data. The main advantage of our approach over a purely function based evaluation is that functional evaluations can only generate a tiny portion of the full problem space by producing variations of the same problem with slightly different numerical values. Their results also suggest substantial amounts of data contamination, including for frontier models, in the MATH dataset.

## 3 GSM1k

GSM1k consists of 1205 problems requiring only elementary mathematical reasoning to solve. We created GSM1k using human annotators. Annotators were prompted with 3 example GSM8k problems and asked to produce novel problems of a similar difficulty level. The precise instructions and UI given to the annotators is available in Appendix A. All problem annotators were instructed to create problems solvable with only basic arithmetic (addition, subtraction, multiplication, and division) and which did not require any advanced math concepts. As is the case with GSM8k, all problem solutions are positive integers[2]. No language models were used to construct this dataset.

To prevent data contamination concerns with GSM1k, we do not intend to release the dataset publicly at this time. However, we commit to releasing the full GSM1k dataset when at least one of the two following conditions have passed, whichever comes earlier. 1) Three open-source models with different pre-trained foundational model lineages reach 95% accuracy on GSM1k. 2) June 2025. At such a point, we believe that grade school mathematics will likely no longer be difficult enough to materially benchmark model releases and commit to releasing all data into the public domain under the MIT license. Additionally, to evaluate proprietary models, we were required to send over the dataset via API. Our belief is that model providers typically do not use such datapoints for model training. Nevertheless, in case GSM1k data is leaked through such means, we also hold out a small number of data points that have passed all quality checks but do not appear in the final GSM1k dataset. This data will also be released alongside GSM1k upon final release. We encourage future benchmarks to follow a similar pattern, where they are not released publicly lest they be gamed, but are precommitted to be released at a future date or upon a future condition. As part of this release, we will also open source our evaluation framework, which is based off of a fork of the LM Evaluation Harness by EleutherAI (Gao et al. [2023a]).

Finally, while we undertook extensive efforts to ensure maximum similarity between GSM8k and GSM1k, these results are only an approximation of an ideal world in which the test set of GSM8k was instead not publicly released and used for evaluations. We would recommend reading all results with the understanding that GSM8k and GSM1k are only highly similar, but not identically distributed despite all our efforts below.

### 3.1 Quality Checks

All questions passed through a total of 3 review layers. After initial creation, each task was manually reviewed by a subset of trusted annotators selected for strong past performance. These reviewers checked both for correctness as well as ensuring problems contained only grade school level math and proper formatting. To ensure that questions were answered correctly, we also do a second review layer by having an independent set of data annotators solve each question *without seeing the intended solution*. If this second solve produced a different answer to that of the initial solve, we discarded the problem. Finally, all problems were reviewed by a special team within Scale responsible for conducting general quality audits for data production. Out of a total of 2108 initial problems, 1419 passed the second solve stage and 1375 passed the general quality audit.

### 3.2 Matching the Difficulty Distribution of GSM8k

One important axis of recreating a benchmark is ensuring that new problems have a comparable difficulty to the original benchmark. To construct problems of difficulty $N$, we requested annotators to construct problems with $N$ required resolution steps and prompted them with 3 examples from GSM8k with estimated difficulty $N$. The distribution of problems requested from annotators matched the estimated distribution in GSM8k. Difficulty is tricky to measure precisely, so we used an estimate based on the number of operations needed to solve the problem. This was extracted programmatically by counting the number of "calculator" tags in the problem solution. However, as not all problem solutions were formatted consistently, this estimate is only a rough estimate of actual difficulty.

---

[2]GSM8k has a few problems, likely errors, for which this is not the case.

Additionally, the number of resolution steps in a problem does not necessarily directly correlate with the true level of problem difficulty.

Past work has also found that LLMs struggle with problems with larger numbers (Gao et al. [2023b]) even if they can solve otherwise identical problems with smaller numbers. To remove this as a potential confounding variable, our final processing step is to discard candidate problems from GSM1k so that the answer magnitude distributions of GSM8k and GSM1k are as similar as possible. This selection process is described in Figure 3. GSM1k consists of the 1205 problems that survive this final winnowing. Additionally, we run several checks to ensure that our efforts to match benchmark

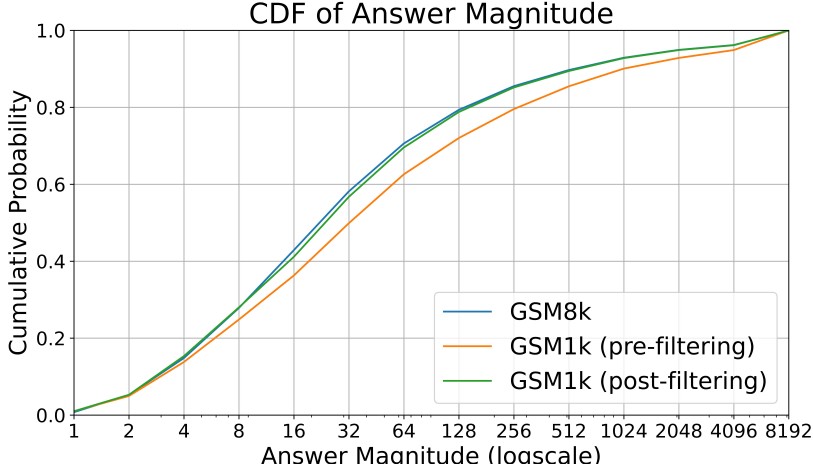

Figure 3: As the final step, we select 1205 problems to match the answer magnitude distribution of GSM8k as much as possible. The remaining problems are discarded and not included in the final dataset. Before discarding, we find that our generated problems tend to have slightly larger answers.

difficulty were successful.

### 3.2.1 Human Differentiation Rates

The first test we run is human distinguishability. We present human annotators with a set of five questions, four of which were randomly selected from the original GSM8k dataset and one of which was selected from the newly created GSM1k dataset, and rewarded annotators for finding the odd one out. In an audit conducted using 19 annotators who were not involved in the problem creation process, we found that annotators were able to correctly identify the lone GSM1k example 21.83% of the time out of 1205 attempts (20% is pure chance). Separately, we also tested several paper authors who had not yet seen the data and they were also unable to perform much better than random. This suggests minimal differences between GSM8k and GSM1k, at least as measured by the human eye.

### 3.2.2 Human Solve Rates

To ensure similar solve rates, we also asked annotators to solve questions under time pressure. 14 annotators who had not participated in the problem creation process attempted to solve as many GSM8k problems as they could in 15 minutes and were rewarded based on the number of problems they solved. We repeated this exact setup for GSM1k. Annotators were able to solve an average of $4.07 \pm 0.93$ problems on the GSM8k dataset. They were able to solve $4.36 \pm 1.11$ problems on the GSM1k dataset, where the error rates are the standard deviations of the evaluation. This suggests that GSM1k is comparable in difficulty (and perhaps even slightly easier) than GSM8k. As such, substantial decreases in model accuracy on GSM1k compared to GSM8k are likely not explainable due to differences in dataset difficulty.

### 3.2.3 LLM Solve Rates

Finally, we sanity check our results by measuring solve rates of several models that are known to not be contaminated by GSM8k due to being released before the publication of the GSM8k dataset. Due to the relative scarcity of LLMs trained only on pre-2021 data, we evaluate only GPT-NeoX-20B (Black et al. [2022]) and GPT-2 (Radford et al. [2019]). For these two language models, we find minimal difference between their solve rates of GSM8k and GSM1k (Figure 12).

## 4 Results

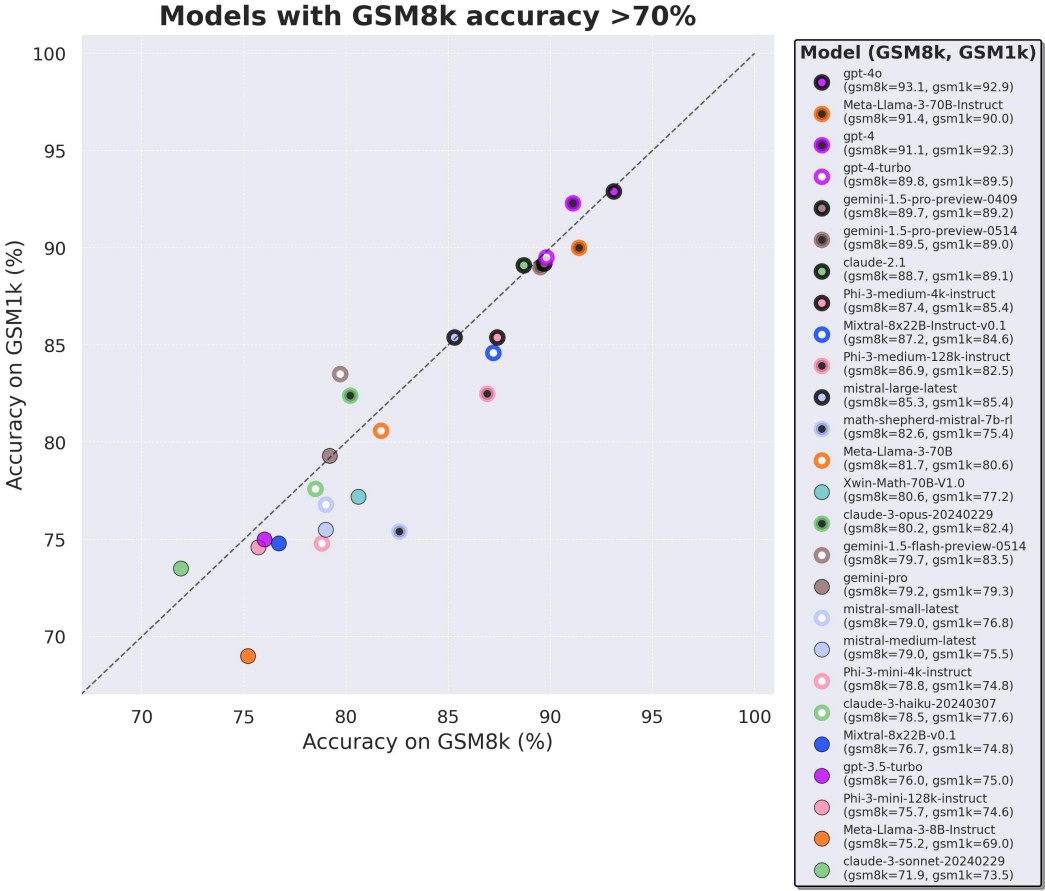

Figure 4: Models with over 70% accuracy on GSM8k compared to the line of no overfit. This plot is zoomed into the relevant sections (70-100% accuracy). Note that some models, especially the Claude family, perform above the 45 degree line, which is consistent with our findings in Section 3 that GSM1k is slightly easier than GSM8k. In contrast, many other models lie well below this line.

To evaluate models, we use a fork of EleutherAI's LM Evaluation Harness with minor modifications. We use the default settings for evaluation, except for increasing the maximum number of allowed generated tokens from 256 to 1000, as we notice that the default setting did not allow some models to complete their full chain-of-thought reasoning before being truncated. Both GSM8k and GSM1k questions are run with the same prompt of using 5 randomly drawn examples from the GSM8k train set, as is standard in the field. An example prompt is provided in Appendix B. All open-source models are evaluated at temperature 0 for reproducibility. For open source models, we use vLLM to speed up model inference if a model is compatible with the library. Otherwise, we default to inference using standard HuggingFace libraries. Closed-source models were queried through the LiteLLM library

which unifies the API call format for all proprietary models evaluated. All API model results were from queries between April 16 - June 5, 2024 and use the default settings.

LM Evaluation Harness uses an automatic evaluation method which extracts the last numeric answer in the response and compares this to the correct answer. However, in some cases, models will produce "correct" answers in a format that do not match the given examples, resulting in their answers being marked as incorrect. To explore the effect of this on the results, we run an ablation where we select a subset of models and use human annotation to manually extract answers that are not correctly formatted (Appendix H). We do not find major changes in our findings for the models examined.

As model benchmark performance is highly dependent on choice of prompt and evaluation setting, our reported GSM8k numbers may occasionally be below the reported model benchmark numbers, as we use a standardized setting for all models instead of the prompt that maximizes each individual model's performance. Additionally, we explore the effect of different prompt formulations with several ablations. In Appendix C, we report results with an alternative prompting format that uses non-GSM8k examples as n-shot examples and a slightly different answer phrasing. Additionally, we explore the effect of varying the number and source of the n-shot examples used in Appendix I and J. While the precise benchmark accuracies vary depending on the setup, we find that the general trends of overfitting hold consistently across our ablations. We will release the full evaluation code for transparency.

In addition to evaluating widely known models, we additionally evaluate several lesser known models that sit near the top of the OpenLLMLeaderboard and discover evidence of Goodhart's law: many of these models perform substantially worse on GSM1k, suggesting that they are primarily gaming the GSM8k benchmark rather than improving model reasoning capabilities. The full set of results, including the performance table for all models, can be found in Appendix D. For fair comparison, we partition the models by performance on GSM8k and compare them to other models which perform similarly (Figures 4, 11, 12).

## 5   Analysis

The interpretation of evaluation results, like the interpretations of dreams, is often a very subjective endeavor. While we report our objective results in Section 4 and Appendix D, here we describe four major takeaways from interpreting the results in a more subjective manner.

### 5.1   Lesson 1: Some Model Families are Systematically Overfit

While it is difficult to draw conclusions from singular data points or model releases, examining a family of models and observing a pattern of overfitting enables us to make more definitive statements. Several families of models, including the Phi and Mistral families of models, show systematic tendencies to perform stronger on GSM8k compared to GSM1k for almost every release and scale of models. Other model families, such as Yi, Xwin, Gemma and CodeLlama also show this pattern to a lesser extent.

### 5.2   Lesson 2: Other Models, Especially Frontier Models, Show No Signs of Overfitting

Nevertheless, we find that many models, through all regions of performance, show minimal signs of being overfit. In particular, we find that all frontier or close-to-frontier models (including the proprietary Mistral Large) appear to perform similarly on both GSM8k and GSM1k. We posit two potential hypotheses for this: 1) frontier models have sufficiently advanced reasoning capability so that they can generalize to new problems even if they have already seen GSM8k problems in their training set, 2) frontier model builders may be more careful about data contamination.

While it is impossible to know for certain without looking at the training set for each model, one piece of evidence in favor of the former is that Mistral Large is the *only* model in the Mistral family to show no signs of overfitting. Since the hypothesis that Mistral took unique care in ensuring that

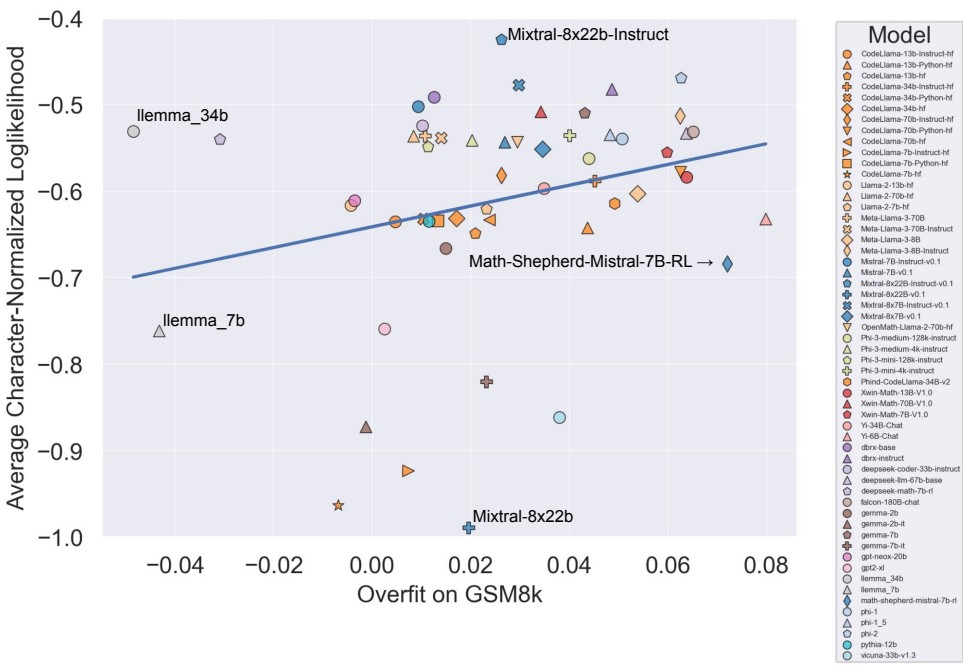

Figure 5: Comparison between overfit on GSM8k (x-axis) and average sequence-level log-likelihood on the GSM8k test set (y-axis). We find that there is a correlation between overfit on GSM8k and sequence-level log-likelihood, suggesting that, in general, models that have a high overfit generally have a higher probability of generating the test set. This suggests that some of the GSM8k test set may have leaked into the model training data. The line of best fit is in blue. Additionally, we highlight 5 "outlier" models which we discuss further with Lesson 4.

only their largest model was free from data contamination seems unlikely, we lean instead towards the hypothesis that sufficiently strong LLMs also learn elementary reasoning ability during training. If a model learns strong enough reasoning capabilities to solve problems of a given difficulty, it will be able to generalize to new problems even if GSM8k has appeared in its training set.

## 5.3 Lesson 3: Overfit Models Are Still Capable of Reasoning

One worry about model overfitting is that models are incapable of reasoning and are only memorizing answers seen in the training data. Our results do not support this conjecture. The fact that a model is overfit does not mean that it is poor at reasoning, merely that it is not as good as the benchmarks might indicate it to be. In fact, we find that many of the most overfit models are still capable of reasoning and solving novel problems. For example, while Phi-2 has a 6% drop in accuracy between GSM8k and GSM1k, we find that it is still able to correctly solve over half of GSM1k problems – which are certain to not have appeared in its training distribution. This performance is similar to that of much larger models such as Llama2-70B, which contains over 25x as many parameters. Similarly, Mistral models remain some of the strongest open source models, even accounting for their overfitting. This provides additional evidence for our lesson that sufficiently strong models learn elementary reasoning, even if benchmark data accidentally leaked into the training distribution, as is likely to be the case for the most overfit models.

## 5.4 Lesson 4: Data Contamination Is Likely Not The Full Explanation for Overfitting

A priori, a natural hypothesis is that the primary cause for overfitting is data contamination, e.g. that the test set was leaked in the pre-training or instruction fine-tuning part of the model creation. Previous work has suggested that models put higher log-likelihoods on data that they have seen

during training (Carlini et al. [2023]). We test the hypothesis that data contamination is the cause of overfitting by measuring a model's probability of generating an example from the GSM8k test set and comparing it to how overfit it is on GSM8k vs GSM1k, using the assumption that a model's probability of generating the GSM8k test set is a proxy for whether the sequence is likely to have appeared in the training set. We normalize by $c$, the number of characters in the sequence, to make the log-likelihood calculations comparable between sequences and models with different tokenizers. Formally, we have:

$$\frac{1}{c} \sum_i \log p(x_i | x_{<i}) \tag{1}$$

with $c$ being the number of characters in the sequence. Figure 5 shows the result of this plot against the gap between GSM8k and GSM1k performance. We indeed find a positive relationship between the two values. We observe a Spearman's rank correlation of 0.36 between the per-character log-likelihood of generating GSM8k and the performance gap between GSM8k and GSM1k ($p = 0.03$), and the relationship suggests that every percentage point difference in GSM8k and GSM1k performance is associated with an increase of $1.2 \times 10^{-2}$ in the per-character log-likelihood. This result suggests that some of the reason for overfitting is due to partial memorization of the test set. For completeness, we also report the standard Pearson $r^2 = 0.26$ and the Kendall's $\tau$ of 0.29, but note that Pearson $r^2$ is not the ideal metric due to the curve-of-best-fit not appearing linear.

Nevertheless, data contamination is likely not the full story. We observe this via the presence of several outliers, which cause the $r^2 = 0.36$ value to be relatively low. Examining these outliers carefully reveals that the model with the lowest per-character log-likelihood (Mixtral-8x22b) and the model with the highest per-character log-likelihood (Mixtral-8x22b-Instruct) are not only variations of the same model, but also have similar levels of overfit (Jiang et al. [2024]). Perhaps more intriguingly, one the most overfit models we discovered (Math-Shepherd-Mistral-7B-RL (Yu et al. [2023])) had a relatively low per-character log-likelihood. Math Shepherd trains a reward model on process level data using synthetic data. As such, we hypothesize that the reward modelling process may have leaked information about the correct reasoning chains for GSM8k even if the problems themselves did not ever appear in the dataset. Finally, we observe that the Llema models (Azerbayev et al. [2024]) have both high log-likelihoods and minimal overfit. These models are open-sourced alongside their training data, and the authors report finding a very small number of GSM8k examples in the training corpus. Nevertheless, they also find (and our study supports) that these few instances do not lead to overfitting. The existence of these outliers suggests that overfitting on GSM8k is not purely due to data contamination, but rather may be through other indirect means, such as model builders collecting data similar in nature to benchmarks as training data or selecting final model checkpoints based on performance on benchmarks, even if the model itself may have not seen the GSM8k dataset at any point via training. Conversely, the reverse is also true: small amounts of data contamination do not necessarily lead to overfitting.

## 6  Discussion

We create GSM1k, a novel dataset designed to measure LLM overfitting on GSM8k. When benchmarking leading open- and closed-source models, we find substantial evidence that many models have been contaminated by benchmark data, with models showing performance drops of up to 8% accuracy. Additionally, we find that several model families show consistent overfitting across almost all model sizes and versions. An extended analysis reveals a positive relationship between a model's likelihood of generating data points in GSM8k and its performance difference between GSM8k and GSM1k, suggesting evidence of data contamination as one of the underlying causes. Nevertheless, we find that frontier models exhibit little to no evidence of overfitting and that many models, even the most heavily overfit families, show strong signs of generalizable mathematical reasoning.

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
