# OpenReview forum: "A Careful Examination of Large Language Model Performance on Grade School Arithmetic"
_NeurIPS.cc/2024/Datasets_and_Benchmarks_Track — NeurIPS 2024 Track Datasets and Benchmarks Spotlight_

### Official Review · Reviewer_7fg6 · 2024-06-24
**An interesting paper**

**Rating:** 7
**Confidence:** 4
**Correctness:** Yes
**Clarity:** Yes

**Review:**

Pros:
- Novel benchmark design: The creation of GSM1k as a control dataset for GSM8k is a novel and valuable approach to evaluating LLM reasoning abilities and identifying potential overfitting.
- Rigorous dataset construction: The paper describes meticulous efforts to ensure the similarity of GSM1k to GSM8k in terms of difficulty, human solve rates, and other metrics, allowing for a fair comparison.
- Detailed analysis and insights: The paper offers a nuanced analysis of the results, identifying patterns of overfitting across model families and exploring the implications for LLM development and evaluation.

Cons:

- Generalizability concerns: The findings are primarily focused on GSM8k and GSM1k, and it’s unclear how well they generalize to other reasoning tasks or domains.

- Limited Scope of Rephrasing Techniques: The paper primarily focuses on the costly human annotation as rephrasing techniques to create GSM1K. However, there may be other methods, such as paraphrasing, translation and altering the context, could these methods lead to similar rephrasing results?

**Strengths:**

- Comprehensive model evaluation: The paper evaluates a wide range of leading open- and closed-source LLMs, providing a comprehensive view of the landscape and highlighting the prevalence of overfitting.

- Thoughtful discussion of overfitting causes: The paper goes beyond data contamination and considers other potential causes of overfitting, contributing to a deeper understanding of the issue.

**Additional Feedback:**

N/A

**Documentation:**

Yes

**Limitations:**

Yes

**Opportunities For Improvement:**

- Limited scope of overfitting analysis: While the paper explores overfitting on GSM8k, it doesn’t fully address other potential benchmarks or datasets where overfitting might occur.
- Can LLMs learn from rephrased samples? The paper focuses on the issue of over-estimation on GSM8K benchmark. However, if an LLM trained on extensive rephrased samples of GSM8K really learned how to solve GSM problems, will the benchmark decontamination (GSM1K) still be necessary? I hope authors can discuss more on the potential pros and cons of training on rephrased samples [1,2].

[1] Dekoninck, Jasper, et al. "Evading Data Contamination Detection for Language Models is (too) Easy." arXiv preprint arXiv:2402.02823 (2024).

[2] Tu, Shangqing, et al. "DICE: Detecting In-distribution Contamination in LLM's Fine-tuning Phase for Math Reasoning." arXiv preprint arXiv:2406.04197 (2024).

**Relation To Prior Work:**

Yes

**Summary And Contributions:**

This paper investigates the data contamination of large language models (LLMs) by creating a new dataset, GSM1k, specifically designed to assess potential overfitting on the existing GSM8k benchmark. GSM1k aims to replicate the style and complexity of GSM8k while ensuring it doesn’t leak into training data. Overall, the paper highlights the importance of carefully evaluating LLM reasoning capabilities and the need for more robust benchmarks to avoid overfitting and accurately assess model performance.

---

> ### Author Rebuttal · Authors · 2024-08-16
>
> Thank you for your comments and suggestions!
>
> Regarding generalizability. We are also very curious about whether these results will generalize to other datasets beyond GSM8k. We hope that other members of the community can expand on this direction and are currently ourselves in the process of doing so for harder math datasets.
>
> Regarding rephrasing techniques for creating GSM1k. As an early initial experiment, we tried using LLM’s to rephrase questions and replace the numbers in each question. However, we ran into challenges in ensuring accuracy of the rephrased problem. Additionally, we were concerned that this method would provide an unfair advantage to the LLM that was used to create the rephrasing. As such, since these rephrased questions would likely have required human quality audits regardless, we decided to simply create the problems from scratch to avoid any such issues with the resulting analysis.
>
> Regarding rephrasing techniques during training. We view GSM1k as a (more expensive) final check for contamination as compared to the methods described in your linked papers. While rephrasing training data might evade checks based on embedding similarity or n-gram overlap, it is unlikely to score well on GSM1k (which is held out) unless it has learned some form of elementary mathematical "reasoning.”

---

### Official Review · Reviewer_KwgF · 2024-07-20

**Rating:** 8
**Confidence:** 4
**Correctness:** Yes, I think that's generally correct.
**Clarity:** Yes, I think the paper writing was ge…

**Review:**

### Pros

- The issue of data contamination is indeed becoming increasingly severe. I have consistently believed that we need trustworthy organizations to conduct fair evaluations without releasing the test data. Instead, they can only provide a few examples to give the research community a general idea of the dataset's content. The introduction of this benchmark is very timely.
- The paper explores the relationship between a model's likelihood of generating GSM8k examples and its performance gap between the two benchmarks, indicating partial memorization of GSM8k. I believe this analysis is highly reasonable.

### Cons
- I believe the research community is likely to be concerned with the performance of their models on GSM1k. Have you considered providing a testing interface (such as specifying the names of models on Hugging Face) to facilitate evaluation by the research community? Additionally, imposing a limit on the number of submissions may be an effective measure to prevent potential overfitting caused by repeated submissions.

**Strengths:**

- The issue of data contamination is indeed becoming increasingly severe. I have consistently believed that we need trustworthy organizations to conduct fair evaluations without releasing the test data. Instead, they can only provide a few examples to give the research community a general idea of the dataset's content. The introduction of this benchmark is very timely.
- The paper explores the relationship between a model's likelihood of generating GSM8k examples and its performance gap between the two benchmarks, indicating partial memorization of GSM8k. I believe this analysis is highly reasonable.

**Additional Feedback:**

No more comments.

**Documentation:**

The authors have released a portion of the testing code. However, they have indicated that the dataset will not be made publicly available at this time.

**Ethics:**

No.

**Limitations:**

No, they do not appear to have articulated the potential limitations.

**Opportunities For Improvement:**

- I believe the research community is likely to be concerned with the performance of their models on GSM1k. Have you considered providing a testing interface (such as specifying the names of models on Hugging Face) to facilitate evaluation by the research community? Additionally, imposing a limit on the number of submissions may be an effective measure to prevent potential overfitting caused by repeated submissions.

**Relation To Prior Work:**

Yes.

**Summary And Contributions:**

The paper presents an in-depth investigation into the performance of LLMs on mathematical reasoning tasks, specifically focusing on the potential issue of dataset contamination. The authors created a new benchmark GSM1k, designed to mirror the GSM8k, to assess whether high performance on the latter might be attributed to models memorizing dataset examples rather than good reasoning capabilities. The study found significant drops in accuracy when evaluating leading LLMs on GSM1k. Notably, some model families exhibited systematic overfitting to GSM8k, while most frontier models still showed minimal signs of such issues.

---

> ### Author Rebuttal · Authors · 2024-08-16
>
> Thank you for your thoughtful review. We are in fact in the process of working with HuggingFace on collaborating to create an open leaderboard for such models! We are hoping to have this out in the next few months and are mostly waiting for some of the other organizations who are also in this collaboration to finish up creating their datasets!

---

> > ### Comment · Reviewer_KwgF · 2024-08-31
> >
> > Thanks for the clarification. I have no more concerns and look forward to the leaderboard of GSM1K.

---

### Official Review · Reviewer_adMu · 2024-07-28
**Initial review**

**Rating:** 9
**Confidence:** 4
**Correctness:** I didn't identify any flaws regarding…
**Clarity:** The paper is well organized and easy …

**Review:**

Pros:

- Data contamination is a big concern and adds noise to understand LLM's real ability. This paper takes an awesome initiate to clear the fog in evaluating frontier LLM's reasoning capability. It is a much needed work and I believe it would be very impactful.
- The entire benchmark is annotated by human, and went through a very complementary and strict process to ensure it is similar to GSM8K.
- The analysis in experiments section is very cautious, especially those on Mistral Large vs. other Mistral smaller models, makes good sense to me.


Cons:

-This is great and much needed work. I do not have nitpicking concerns.

Questions:

- I'm not totally sold by the lesson 3 in experiment chapter. The authors demonstrate reasoning capability of overfit models by examples " Phi-2  is better than llama2-70B in GSM1K". Since GSM8K is likely included in training of Phi family models, is it possible its advantage over llama2-70 comes from more GSM style questions in training data? In other words, if a model is trained on more GSM style data, it is not surprised if they're better at similar new GSM questions. But this does not mean they have better reasoning capability.

- GSM1K aims to be a private benchmark to avoid data contamination. However, the questions can still be accessed if run APIs of closed models. Though those big names might not misuse them, I'm still curious if the authors have any strategy that can prevent the leak of private data. Perhaps updating the benchmark in a periodic manner would help (cost++ though)?

- Purely curious question: since the entire dataset includes tons of human annotator efforts, if possible, could the authors introduce an estimate of time and cost in building this benchmark?

**Strengths:**

I feel this section is sort of repeated,  please just check the Review section.

**Additional Feedback:**

N/A

**Documentation:**

This paper present a new dataset - the data creation process is well discussed.

**Opportunities For Improvement:**

Again I feel this section is sort of repeated,  please just check the Review section.

**Relation To Prior Work:**

Prior work such as previous elementary-level math benchmarks, and previous efforts on detecting/preventing data contamination are both well discussed.

**Summary And Contributions:**

This paper creates a new elementary math benchmark GSM1K to measure how LLMs overfits the popular GSM8K benchmark. GSM1K goes through a very strict process to ensure its questions matches GSM8K in every possible perspective (style, difficulty, etc). Running LLMs on GSM1K shows that some family of models (e.g., Phi, mistral) are likely to have data contamination issues, while frontier models such as GPT-4 and Claude show no overfitting.

---

> ### Author Rebuttal · Authors · 2024-08-16
>
> Thank you for your kind comments! Regarding Llama2-70B vs Phi-2, we agree with your statement and will rephrase to specifically reference reasoning in elementary mathematics. Our intention was to convey that, despite being overfit, Phi-2 still generalizes to solving GSM1k roughly on the level of Llama2 70B, showing that its performance, while affected by data contamination, cannot be entirely due to memorization.
>
> Regarding contamination of GSM1k, we largely trust that closed model APIs will be careful with data sent to them. That said, we are also currently in the process of creating a refreshed, harder math benchmark to evaluate the next generation of LLMs.
>
> Regarding cost. In Appendix B, we include some additional details related to the dataset creation, including cost. In total, GSM1k cost around 180K USD, with each data annotator being paid between 20-25 dollars an hour.

---

### Comment · Area_Chair_nDD1 · 2024-08-29
**Please engage in the author-reviewer discussion**

Dear Reviewers,

Thank you for your hard work on the papers and reviews. Please note that the deadline for the author-reviewer discussion period is approaching (August 31, 2024). Some of you have engaged in discussions with authors-thank you! For reviewers who have not yet, please discuss with authors as this is a very important part of the reviewing process and authors are eager to have further feedback from you. If there are any changes to your scores, kindly provide explanations for these adjustments.

---

### Decision · Program_Chairs · 2024-09-26

**Decision:**

Accept (Spotlight)

**Comment:**

The paper creates a held out version of GSM8k with a focus on dataset contamination.  With this dataset, the paper presents an in-depth investigation into the performance of LLMs on mathematical reasoning tasks.

Pros:
1. The paper is well-motivated, investigating an important evaluation issue data contamination.
2. The paper presents a rigorous dataset annotation process.
3. The paper provides cautious analysis and insights.